# Design of the HPV-automated visual evaluation (PAVE) study: Validating a novel cervical screening strategy

Silvia de Sanjosé[1,2]*, Rebecca B Perkins[3], Nicole Campos[4], Federica Inturrisi[1], Didem Egemen[1], Brian Befano[5,6], Ana Cecilia Rodriguez[1], Jose Jerónimo[1], Li C Cheung[1], Kanan Desai[1], Paul Han[7], Akiva P Novetsky[8], Abigail Ukwuani[1], Jenna Marcus[9], Syed Rakin Ahmed[10,11,12,13], Nicolas Wentzensen[1], Jayashree Kalpathy-Cramer[10,14], Mark Schiffman[1], On behalf of the PAVE Study Group

[1]Division of Cancer Epidemiology and Genetics, National Cancer Institute, National Institutes of Health, Rockville, United States; [2]ISGlobal, Barcelona, Spain; [3]University Chobanian and Avedisian School of Medicine/Boston Medical Center, Boston, United States; [4]Center for Health Decision Science, Harvard T.H. Chan School of Public Health, Boston, United States; [5]Information Management Services Inc, Calverton, United States; [6]Department of Epidemiology, University of Washington School of Public Health, Seattle, United States; [7]Division of Cancer Control and Population Sciences, National Cancer Institute, National Institutes of Health, Rockville, United States; [8]Westchester Medical Center/New York Medical College, Valhalla, United States; [9]Feinberg School of Medicine at Northwestern University, Chicago, United States; [10]Athinoula A. Martinos Center for Biomedical Imaging, Department of Radiology, Massachusetts General Hospital, Boston, United States; [11]Harvard Graduate Program in Biophysics, Harvard Medical School, Harvard University, Cambridge, United States; [12]Massachusetts Institute of Technology, Cambridge, United States; [13]Geisel School of Medicine at Dartmouth, Dartmouth College, Hanover, United States; [14]University of Colorado Anschutz Medical Campus, Aurora, United States

*For correspondence: desanjose.silvia@gmail.com

Sent for Review 17 August 2023
Preprint posted 31 August 2023
Reviewed preprint posted 16 October 2023
Reviewed preprint revised 05 December 2023
Version of Record published 15 January 2024

## Abstract

**Background:** The HPV-automated visual evaluation (PAVE) Study is an extensive, multinational initiative designed to advance cervical cancer prevention in resource-constrained regions. Cervical cancer disproportionally affects regions with limited access to preventive measures. PAVE aims to assess a novel screening-triage-treatment strategy integrating self-sampled HPV testing, deep-learning-based automated visual evaluation (AVE), and targeted therapies.

**Methods:** Phase 1 efficacy involves screening up to 100,000 women aged 25–49 across nine countries, using self-collected vaginal samples for hierarchical HPV evaluation: HPV16, else HPV18/45, else HPV31/33/35/52/58, else HPV39/51/56/59/68 else negative. HPV-positive individuals undergo further evaluation, including pelvic exams, cervical imaging, and biopsies. AVE algorithms analyze images, assigning risk scores for precancer, validated against histologic high-grade precancer. Phase 1, however, does not integrate AVE results into patient management, contrasting them with local standard care.

Phase 2 effectiveness focuses on deploying AVE software and HPV genotype data in real-time clinical decision-making, evaluating feasibility, acceptability, cost-effectiveness, and health communication of the PAVE strategy in practice.

**Results:** Currently, sites have commenced fieldwork, and conclusive results are pending.

**Conclusions:** The study aspires to validate a screen-triage-treat protocol utilizing innovative biomarkers to deliver an accurate, feasible, and cost-effective strategy for cervical cancer prevention in resource-limited areas. Should the study validate PAVE, its broader implementation could be recommended, potentially expanding cervical cancer prevention worldwide.

**Funding:** The consortial sites are responsible for their own study costs. Research equipment and supplies, and the NCI-affiliated staff are funded by the National Cancer Institute Intramural Research Program including supplemental funding from the Cancer Cures Moonshot Initiative. No commercial support was obtained. Brian Befano was supported by NCI/ NIH under Grant T32CA09168.

## eLife assessment

This **important** study will provide evidence about a novel screen-triage-treat strategy for cervical cancer prevention. The trial will generate **convincing** evidence regarding the efficacy, effectiveness, cost-effectiveness, feasibility and acceptability in a range of geographically spread low-resource settings. The strategy should contribute to improving access to cervical cancer prevention to vulnerable women with low access to health care, and, therefore, at the highest risk of cervical cancer.

## Introduction

### Global burden of cervical cancer

Cervical cancer causes substantial morbidity and mortality worldwide, with approximately 600,000 incident cases and 340,000 deaths each year (*Singh et al., 2023*). Globally, cervical cancer is caused by persistent infection with one of ~13 carcinogenic human papillomavirus (HPV) types (*Schiffman et al., 2007*). Cervical cancer rates vary greatly worldwide due to uneven access to effective preventive measures; nearly 85% of cervical cancer cases and almost 90% of cervical cancer deaths occur in low- and middle-income countries (LMIC) (*IARC, 2022*; *Ferlay et al., 2020*). The World Health Organization (WHO) has called for the global elimination of cervical cancer, based on an advanced understanding of the natural history of the causal carcinogenic types of cervical HPV infection and the existence of effective preventive technologies, including prophylactic HPV vaccination and cervical screening (*Schiffman et al., 2007*; *World Health Organization, 2021*; *WHO, 2020*). However, translation of the HPV-based prevention methods has not yet occurred in many LMICs.

While prophylactic vaccination will eventually decrease cervical cancer rates (*Lei et al., 2020*) if high uptake can be achieved in LMIC, the maximum potential health benefits of vaccinating adolescents today will not be achieved for 40 years. However, the rapid implementation of a broad, effective cervical screening campaign for adult women in the highest-burden areas will advance cancer control by 20 years (*Figure 1*). The U.S. Cancer Moonshot initiative for Accelerated Control of Cervical Cancer has supported development of the new screening methods (*Perkins et al., 2023*).

### Screening using HPV testing

The detection of carcinogenic cervical/vaginal HPV DNA is currently the most sensitive screening method to distinguish those with an appreciable risk of precancer or cancer from those at low risk (*Demarco et al., 2022*; *Castle et al., 2012*). The WHO currently recommends using either screen-treat or screen-triage-treat strategies. HPV testing is preferred to using visual inspection with acetic acid (VIA) as the primary screening method where resources permit (*World Health Organization, 2021*). There is a growing consensus that to achieve broad screening coverage, HPV testing of self-collected cervicovaginal specimens would be optimal for many populations (*IARC, 2022*; *World Health Organization, 2021*; *Perkins et al., 2023*; *Arbyn et al., 2018*). The results from meta-analyses comparing the performance of self-collected to clinician-collected samples, using PCR-based HPV detection, showed similar sensitivity and specificity for the detection of cervical precancer (*Arbyn et al., 2018*). As of 2022, seven LMICs were recommending HPV self-collection (*Serrano et al., 2022*).

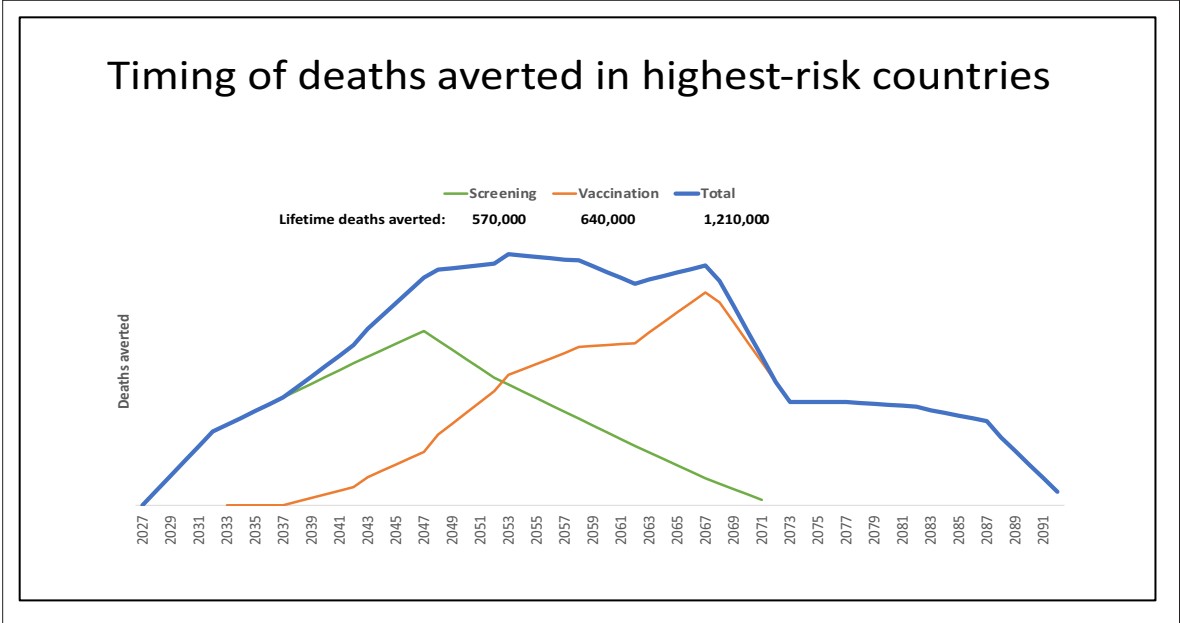

**Figure 1.** Timing and deaths averted with one-time prevention campaigns: vaccination only, screening only, or both. Projection of the relative timing of health benefits, measured as deaths averted, accrued by vaccination and/or screening applied through one-time campaigns. Three scenarios were examined: (1) a one-time screening campaign providing effective management for approximately 25% of 30- to 49-year-old women in 2027 (i.e. 20 birth cohorts) (green line); (2) vaccinating 90% of 9- to 14-year-old girls in 2027 (i.e. six birth cohorts) with a bivalent HPV16/18 vaccination (orange line); and (3) both a screening campaign and human papillomavirus (HPV) vaccination for respective birth cohorts in 2027 (blue line). We considered cervical cancer deaths averted over the lifetime of cohorts subject to the intervention, and conservatively assumed that deaths averted due to screening would only occur after age 50, to account for prevalent cancers. Projections were developed for the ~65 low- and middle-income countries (LMIC) with age-standardized cervical cancer incidence greater than 10 per 100,000 women (**Perkins et al., 2023**). For each country, we assumed that, in the absence of any intervention, the number of cervical cancer deaths for each 5 year age group would apply each year for the lifetime of the selected birth cohorts (**Perkins et al., 2023**). We conservatively and crudely assumed that screening and management would avert 25% of cervical cancer deaths (equivalent to screening uptake of 40% of eligible women, with 62.5% of screen-positive women receiving appropriate management) beginning at age 50 years. For vaccination cohorts, we assumed that a bivalent HPV16/18 vaccine (i.e. against the genotypes responsible for 70% of cervical cancers) with 90% uptake would avert 63% of cervical cancer deaths. While data on the costs of implementing novel screening strategies and single-dose HPV vaccination for female adolescents are forthcoming from the HPV-automated visual evaluation (PAVE) consortium and single-dose vaccination studies, we crudely assumed a single vaccine dose cost US$4.50[31] with an average financial delivery cost per dose (i.e. per fully immunized girl) of US$7 (**Akumbom et al., 2022**). We assumed a bundled financial cost per woman screened of US$15, including a low-cost rapid HPV genotyping assay with triage and treatment of screen-positive women. According to our projections, the number of interventions needed to avert one cervical cancer death was similar for HPV vaccination and screening (i.e. 278 for HPV vaccination; 293 for screening). A one-time screening campaign for women aged 30–49 years in the selected countries yielded a financial cost of ~US$2.5 billion to avert ~570,000 deaths, or US$4,400 per death averted. On a similar order of magnitude, a one-time single-dose bivalent HPV vaccination campaign of girls aged 9–14 years in the same countries would cost ~US$2.0 billion and avert ~640,000 deaths, or US$3,200 per death averted. Of note, these ballpark estimates are undiscounted and do not account for cancer treatment cost offsets. We also did not consider demographic changes over the lifetime of intervention cohorts, nor did we consider the indirect benefits of vaccination or prevention of other HPV-related cancers.

There is broad scientific agreement that HPV testing is the preferred screening method due to its high negative predictive value and reproducibility. Treatment of all HPV-positive women with thermal ablation (i.e. screen-treat strategy) is an option under current WHO guidelines (**World Health Organization, 2021**); however, this may substantially overtreat infections, only the minority of which would progress to cancer (**Demarco et al., 2020**). To make the best use of limited resources and concentrate on those at the highest risk, triage strategies are critical to determine which HPV-positive women are at higher risk of cervical cancer. Triage with cytology or dual stain, as used in high-resource settings, is unlikely to be a feasible solution in the majority of low-resource settings. Cervical visual examination using visual techniques, including VIA, are often used as triage methods. However, these techniques are subject to human error, have low accuracy for precancer, and require continuous training and quality control measures (**Sankaranarayanan et al., 2003**; **Catarino et al., 2018**). HPV genotyping is a newer, more accurate method of triage, as genotype carcinogenicity varies predictably across populations (**Sankaranarayanan et al., 2009**; **Qiao et al., 2008**). HPV16 is the most carcinogenic,

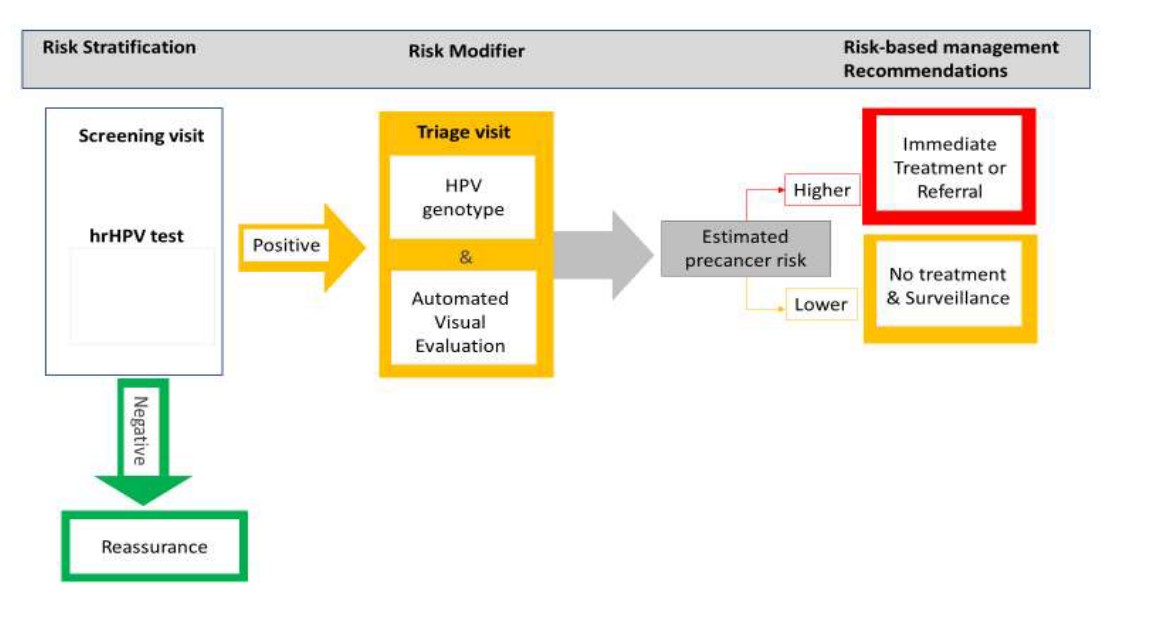

**Figure 2.** Risk-based HPV-automated visual evaluation (PAVE) screen-triage-treat strategy provides risk stratification to assist in the management of screening participants. hrHPV: refers to those human papillomavirus (HPV) types considered as having a high potential capacity to induce cervical cancer when the infection is persistent over time. It includes HPV16, 18, 45, 31, 33, 45, 52, 58, 39, 51, 56, 59, and 68.

followed by HPV18/45, followed by HPV31/33/35/52/58, followed by HPV39/51/56/59/68 (**Guan et al., 2012**). AVE using an Artificial Intelligence (AI) algorithm shows promise as a relatively simple and fast triage method that could be used in conjunction with HPV genotyping to generate a highly accurate composite triage test (**Desai et al., 2020**; **Desai et al., 2022b**).

## HPV-AVE (PAVE) strategy

The U.S. National Cancer Institute (NCI) is currently leading a consortium of a multi-centric study designed to evaluate a novel cervical screening and triage strategy for resource-limited settings, including settings with high HIV prevalence, as part of a global strategy to reduce cervical cancer burden. The PAVE strategy aims to target cervical precancer accurately and affordably by (1) self-sampled HPV screening; (2) triage among HPV-positive participants by combination of extended genotyping and visual evaluation assisted by deep-learning-based AVE; and (3) treatment using thermal ablation or excision Large Loop Excision of the Transformation Zone (LLETZ). PAVE utilizes the concept of risk-based management, defined as patient management determined by their risk of precancer/cancer to minimize overtreatment in low-risk patients and concentrate treatment resources on high-risk patients (**Figure 2**). This manuscript describes the study protocol, structure, and logic of the PAVE strategy.

## Methods

The PAVE study has two phases: efficacy (2023–2024) and effectiveness (planned to begin in 2024 or 2025). The efficacy phase aims to refine and validate the screen-triage portion of the protocol. The effectiveness phase will research the introduction of the PAVE strategy into clinical practice.

### Phase 1: Efficacy
Setting: Study design and locations
The study aims to recruit up to 100,000 women in nine countries: Brazil, Cambodia, Dominican Republic, El Salvador, Eswatini, Honduras, Malawi, Nigeria, and Tanzania (**Figure 3** and **Figure 4**). Criteria for study site selection included: (a) existing screening programs, (b) willingness to research self-sampled HPV for screening, (c) capacity to run the HPV test (d) availability of pathology services

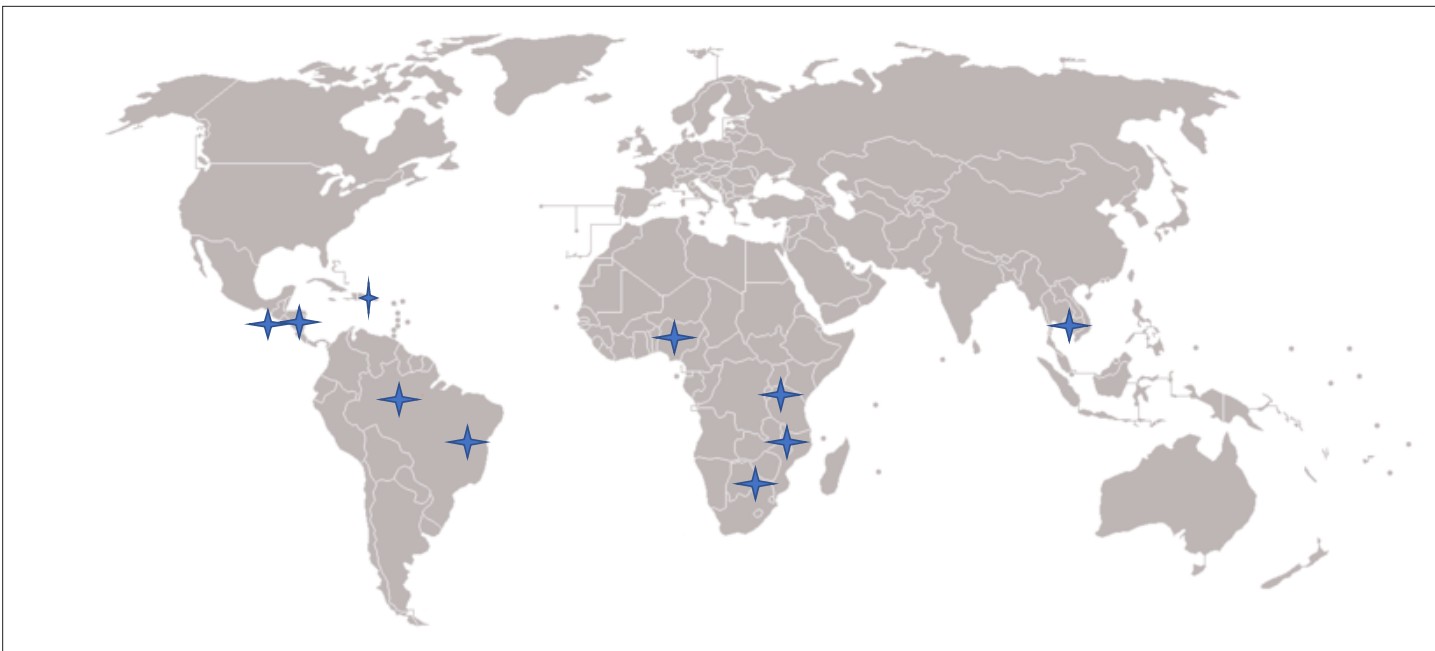

**Figure 3.** Map of HPV-automated visual evaluation (PAVE) study sites.

to process biopsies, and (e) access to treatment services including ablation, excision, and cancer treatment. Outreach and recruitment activities under the protocol include awareness campaigns to inform the eligible female population in the catchment area. Research protocol details including recruitment strategies, number of visits, and institutional review board approval are under the control of individual study sites. The PAVE project is integrated into the screening activities at all study sites, and at select sites is also integrated into other ongoing research studies.

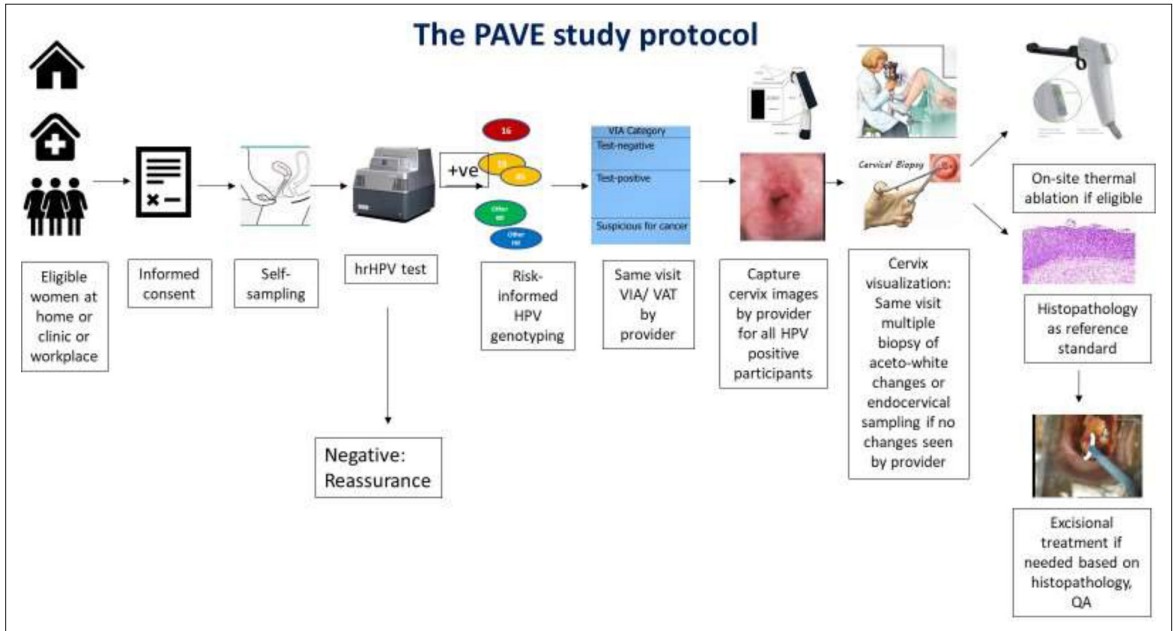

**Figure 4.** Schematic of HPV-automated visual evaluation (PAVE) protocol elements.

## Ethical and regulatory aspects

This multi-centric study is designed to function as a consortium. All ethical oversight of recruitment and clinical data collection will be done by the local sites under the guidance of their own Institutional Review Boards and will follow local guidelines. All participants will have an informed consent for participation in the study and can drop their participation at any time during the process. All study documents written in languages other than English are officially translated into English for study records. Compiled analysis of de-identified data by NCI research staff for study purposes is deemed non-human subjects research by NIH.

Protocol overview (Date of protocol latest review: September 24, 2023).

## In-country elements: Patient enrollment and data collection

The steps of the PAVE protocol include (1) determination of study eligibility, (2) informed consent, (3) self-sampled HPV testing, (4) cervical image collection and biopsy collection for those testing HPV-positive, and (5) treatment as indicated per local protocols (*Figure 4*).

### Enrollment

Eligibility criteria are: individuals with a cervix aged 30–49 years (general population) or 25–49 years if living with HIV (WLHIV), not currently pregnant, and able to understand study risks, benefits, and alternatives and provide informed consent in their native language. Those eligible for and interested in study participation undergo informed consent per local protocols. Those who choose to enroll provide basic demographic information (age, parity, HIV status if known). Pregnant women, or those with no cervix or having been treated for a cervical cancer are excluded.

At each study site, we anticipate ~1–2% prevalence of histopathologic precancers (HPV-positive adjudicated CIN2+/AIS), yielding about 100–200 precancers and a very few cancers out of a sample of 10,000 women. Combined across all countries, we anticipate up to 1000–2000 precancers.

### HPV self-collection

Participants self-collect a vaginal sample for HPV testing using a FLOQSwab (Copan) following instruction by study personnel. Self-samples (FLOQSwab) are delivered dry for testing. All sites intend to use the ScreenFire HPV risk-stratification (RS) assay (ScreenFire) (U.S. patent 11091799, Atila Biosystems Inc, Sunnyvale, CA, U.S.) (*Desai et al., 2022a*; *Inturrisi et al., 2024*). However, other HPV tests may be acceptable alternatives if they can provide genotyping information in the following groups: HPV16, HPV18/45, HPV31/33/35/52/58, and HPV39/51/56/59/68.

HPV tests are run onsite or in local laboratories in a few days, with results returned to women quickly per local protocols. Women screening HPV-negative are informed of their results, reassured about their low subsequent risk of cervical cancer, and their participation in the study ends at most sites. The exception is El Salvador, at which 5% of those screening HPV-negative undergo colposcopic examination. On average, approximately 80% of participants will screen HPV-negative, but this varies by study population.

### Triage of HPV-positive results: Image collection and biopsy

Women with HPV-positive results undergo a speculum exam with an application of 5% acetic acid. Cervical images will be collected by a trained study provider using a dedicated device (Iris, Liger Medical LLC, Lehi, UT, U.S.). Local clinicians also record their VIA assessment (negative, positive, suspicion of cancer) or colposcopy impression (normal, low-grade, high-grade, or more severe).

Following image capture, pathology specimens are collected. Biopsies will be collected from up to four acetowhite areas for each participant. If no acetowhite areas are seen, then an endocervical sample (using curette or brush) or cytology will be collected. Sites using colposcopy will collect punch biopsies per standard practice. Sites using VIA will use Softbiopsy/SoftECC brush biopsy (Histologics LLC, Anaheim, CA, U.S.), a device that is simpler to learn and perform and is associated with lower bleeding risk. All women that have an HPV-positive test are expected to have a histologic diagnosis (biopsy, endocervical sample (ECC), and/or excisional tissue diagnosis). HPV-positive women with a negative triage evaluation initially, but who are later identified by PAVE activities to have a precancer defined for study purposes as HPV-positive CIN2, CIN3 or AIS CIN2+, will be flagged and the clinical sites will be notified to permit 'safety net' recall for adequate management.

**Table 1.** Site-specific primary triage and treatment protocols Biopsy and treatment protocols.

| Site | Primary Screening Test | Triage method | Staff taking biopsies | Biopsy Instrument | Treatment threshold | Primary Treatment |
|------|------------------------|---------------|-----------------------|-------------------|---------------------|-------------------|
| Dominican Republic | ScreenFire HPV, Cytology | Colposcopy | Gynecologists | Biopsy forceps | CIN2+ +biopsy | Ablation or LLETZ |
| Malawi | ScreenFire HPV | VIA | Nurses | Softbrush | VIA-positive | Ablation or LLETZ |
| Nigeria | ScreenFire HPV | Colposcopy | General Doctors, Gynecologists, GYN oncologists | Biopsy forceps | HPV-positive | Ablation or LLETZ |
| Brazil | ScreenFire HPV | Conventional cytology, liquid-based cytology, and colposcopy | Gynecologists | Biopsy forceps | CIN2 + biopsy or high-grade colpo impression | LLETZ |
| Cambodia | ScreenFire HPV | Colposcopy (mobile colposcope) | Nurse midwives, General Doctors, Gynecologists | Softbrush | VIA-positive or HPV16,18/45 positive | Ablation or LLETZ |
| Eswatini | ScreenFire HPV & VIA | VIA | Nurse midwives | Softbrush | VIA-positive | Ablation |
| El Salvador | ScreenFire HPV, Cytology, CareHPV | Colposcopy and VAT | General Doctors | Biopsy forceps | HPV-positive | Ablation |
| Tanzania | ScreenFire HPV | VIA | Gynecologists, nurses, nurse midwives | Biopsy forceps, Softbrush | VIA-positive | Ablation |
| Honduras | ScreenFire HPV | Colposcopy and VAT | Nurse midwives, General Doctors, | Biopsy forceps | VIA-positive | Ablation |

## Treatment

Treatment is provided for women meeting criteria per local protocols: VIA-positive or suspicion of cancer in sites using VIA, CIN2 + on biopsy and/or high-grade colposcopy impression in sites using colposcopy/biopsy, or HPV-positive and acetowhite changes or HPV16, 18/45 positive or HPV-positive for sites using screen-treat protocols. For sites using VIA, treatment decisions for those screening VIA-positive will be based on the standard of care, most commonly an adaptation of the WHO visual assessment for treatment (VAT) criteria for the use of ablation. For sites using LLETZ, treatment decisions will follow local protocols. *Table 1* describes screening, triage, biopsy, and treatment protocols for each site.

Note to *Table 1*: Prior to the PAVE study, El Salvador screened with primary HPV screening (clinician-collected CareHPV), Brazil and DR screened with cytology, and Cambodia, Eswatini, Honduras, Malawi, Nigeria, and Tanzania screened with VIA. All sites are introducing self-sampled HPV testing with ScreenFire as part of the PAVE protocol. In El Salvador, women are continuing to screen initially with both ScreenFire and CareHPV. Triage testing is performed in all women with HPV-positive results. In El Salvador, triage testing is also performed on 5% of those testing HPV-negative.

## Central elements: Data management, HPV testing, AVE algorithm development, quality assurance, statistical analysis

### Data management

Study data including demographic survey information, HPV test results (negative/positive, genotype for positive results), VIA or colposcopy impression, and local pathology results (cytology, biopsy, and/or excisional specimen) are collected using District Health Information System 2 (DHIS2), Redcap or WEMA platforms. The collected data are associated with the corresponding images obtained during the triage visits. To ensure confidentiality, all personal identification information is removed from datasets before transferring outside the country of origin. De-identified datasets are securely transferred to a common server handled by the NGO specialized in country adaptation of DHIS2 information systems Enlace Hispano Americano de Salud (EHAS) and from there, compiled data are transferred to Information Management Services (IMS), the NCI data management contractor, for data storage

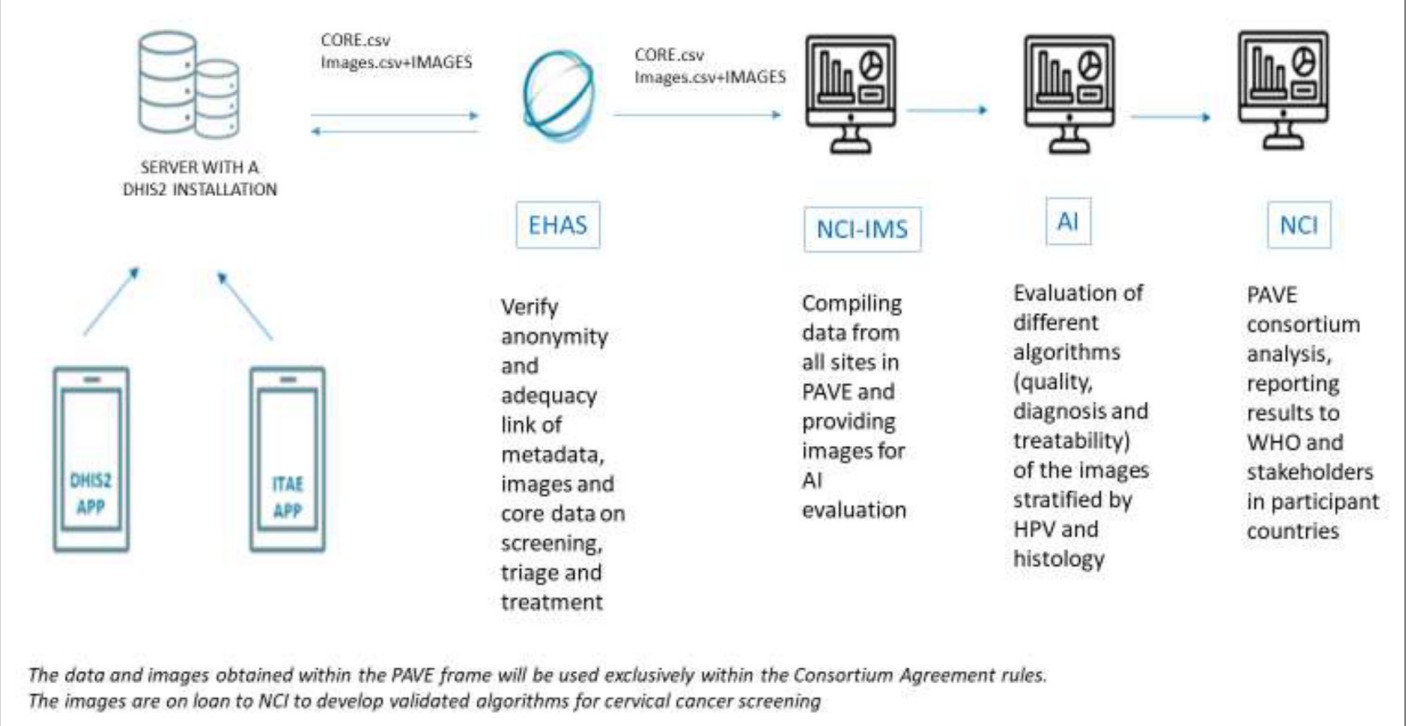

**Figure 5.** Flowchart of data sharing in the HPV-automated visual evaluation (PAVE) study sites using the DHIS2 during the efficacy phase.

and analytic support during the course of the study. The noteworthy element in this study design element is that data rights (and residual biospecimens) will remain with the study site partners within their countries. The data on loan for PAVE analyses will be stored securely and data can be withdrawn and destroyed at any time by study site partners. This arrangement is important to many aspects of international data and biospecimen sharing.

*Figure 5* illustrates the various steps involved in the PAVE consortium, from data collection to data analysis. This outline is for those sites utilizing the DHIS2 app, where pre-defined data is collected from women who agree to participate. It is important to note that any individual information collected for clinical management purposes will not be uploaded to the study server.

The study server incorporates the ITEA app, which facilitates the transfer of data to the NGO EHAS. At EHAS, the data undergoes a thorough check to ensure appropriate study identification. Additionally, if necessary, cervical images of the participants are linked to the core database. Once the verification process is approved, the data is sent to the National Cancer Institute- Information Management System (NCI-IMS) for preparation before analysis.

During this step, data verification and cleaning procedures are performed to ensure the accuracy and quality of the data. Subsequently, the images, as well as the HPV and histology assessments (when available), are forwarded to Prof Jayashree Kalpathy-Cramer's team for AI evaluation. The results, in the form of data scores, are then transmitted back to NCI-IMS for comprehensive analysis of the consortium data.

Note: Sites using other apps like RedCap will extract the Core Data, that includes the information listed at the beginning of this section from their files, and forward it to EHAS. Images will follow the same procedure.

## HPV testing/typing

The ScreenFire HPV test is a new assay designed to detect the 13 high-risk HPV (hrHPV) genotypes grouped into the four risk groups described above, and specifically engineered to provide risk stratification by HPV genotype based on carcinogenicity. ScreenFire includes an internal control for sample quality guidance. The ScreenFire HPV assay is an isothermal, multiplex nucleic acid amplification method that uses 3NT technology to reduce false positivity and increase assay performance.

ScreenFire can be run on 1–96 samples per batch, requires basic pipetting skills, and takes approximately 2.5–3 hr in total including sample preparation, pipetting, and DNA amplification with readout.

## Validation

ScreenFire was compared against reference research HPV DNA assays in 2078 stored specimens. Overall concordance for both viral types was >90%, and sensitivity for CIN3 + was 94.2%, similar to Linear Array (93.1%) and TypeSeq (95.9%), indicating excellent performance (*Inturrisi et al., 2024*). Regulatory approval will require additional comparisons in screening settings. Additional clinical studies will be nested within the PAVE protocol. In El Salvador, 5% of HPV-negative women will undergo colposcopic evaluation. Comparison of ScreenFire to other WHO pre-qualified HPV tests (CareHPV and Abbott) may also be performed on subsets of patients.

## Development of AVE algorithm using clinical images

Cervical images will be transferred via the digital camera to a secure server using a specially designed script. Images and non-PHI data will be shared and downloaded on a loan basis to the PAVE AI team at NCI and the AI collaborators to train the AVE algorithm. Because images from the Iris device have not been used previously with the AI algorithm, a pilot phase will take place to retrain the AI-based algorithm. Due to the similarities of the Iris device to previously tested digital cameras, we expect to be able to retrain the algorithm successfully as done in our prior work (*Parham et al., 2023*).

During the PAVE study, four AI algorithms are being developed and evaluated: (1) cervix detector, (2) image quality classifier, (3) disease classifier to identify precancer, and (4) treatability/SCJ classifier.

1. Cervix detector. This algorithm is designed to display a bounding box on the screen, aiding healthcare providers ensure that the cervix is centralized within the image. This feature simplifies the process of locating the cervix within the digital image, enhancing the efficiency and accuracy of image collection.

2. Image quality classifier. This algorithm aims to identify images that may be unsuitable for accurate disease assessment due to factors like obstruction or inadequate visual sharpness. By flagging such images, it helps ensure that only good-quality images are used when training the algorithm. If shown to be useful, the image quality classifier could provide feedback in real-time to clinicians when taking images to ensure adequate image quality.

3. Disease classifier. This algorithm aims to visually distinguish precancerous changes from lesser abnormalities, and classifies cervical images as normal, indeterminate, or precancer+. AVE results are assessed on repeatability (correct classification of replicate images of the same patient) and accuracy (correct classification of AVE based on histopathology, as well as minimization of extreme misclassification of normal as precancer or vice versa). Accuracy is defined as a correct classification of participants to <precancer or precancer+. Precancer + is rigorously defined to include HPV-positive histologic CIN3, AIS, cancer, and CIN2 diagnoses confirmed by expert pathologic review and positive for the eight most carcinogenic HPV genotypes. CIN2 is qualified because although CIN2 is the threshold for treatment in most clinical practices worldwide, it is a less reproducible pathologic diagnosis and may regress without treatment, especially when associated with lower carcinogenicity HPV genotypes (*Lee et al., 2018*; *Kylebäck et al., 2022*). The current prototype algorithm is the result of several years of development. Earlier algorithms were limited by poor repeatability, misclassification including grave errors (i.e. cases with precancer called normal or vice versa), overfitting, and lack of portability (defined as the ability of the algorithm to accurately classify images from different image capture devices and study settings other than the dataset on which it was trained) (*Desai et al., 2022b*). Additional techniques have been applied to develop the prototype version of the AVE algorithm that will be refined and tested in the PAVE study (*Ahmed et al., 2023a*), resulting in improved reliability and consistency of model predictions across repeat images from the same woman (*Lemay et al., 2022*). The disease classifier algorithm is trained using histology as the truth standard for defining the presence or absence of precancer. The outcome definition for the purpose of training a three-class ordinal classification algorithm includes normal (HPV-positive with histology normal, HPV-negative, HPV-negative cervicitis), indeterminate (low-grade HPV-related abnormalities, CIN1), and precancer+, as determined by histology among HPV-positives (defined above). To ensure portability, the algorithm will undergo external validation using datasets distinct from the training set of images. Early data indicate that while our algorithm may function among patients across diverse geographies (*Ahmed et al., 2023b*) a dedicated device may be needed because AVE fails to transfer without retraining

(*Parham et al., 2023*). The images collected in PAVE will be used to refine and externally validate the prototype AVE algorithm (*Lemay et al., 2022*). We test the repeatability, accuracy, and calibration of the model before the algorithm is tested in clinical settings during the effectiveness phase (*Egemen et al., 2023*).

4. Treatability/SCJ classifier. This algorithm is being developed to classify the SCJ as fully visible, partially visible, or not visible, using the expert colposcopic impression as the truth standard. The goal is to assist providers in determining treatment eligibility. SCJ visibility might be critical (i.e. necessary although not sufficient) for eligibility for thermal ablation procedures.

## Pathology quality assurance

Pathology readings are performed locally with centralized quality assurance on a subset of cases. Histotechnology adequacy via slide review from all participating laboratories includes assessment of specimen preparation, staining adequacy, and clarity/readability of scanned images by the assigned referent NCI study pathologist in collaboration with pathologists at each study site. Local pathologists involved in the PAVE project are asked to complete a performance competency review which includes providing diagnoses on 20 standardized cases. Issues with either slide preparation or interpretation were addressed via videoconference between the NCI expert pathologist and local laboratories.

To assure histopathology reading standardization, the following cases will be reviewed by an expert gynecologic pathologist, making use of a Motic whole slide scan review collection and transfer: (1) all cases with histology CIN2 + or high-grade squamous intraepithelial lesion (HSIL); (2) all HPV16 + with <CIN2 pathology; (3) all cases read as precancer + on AVE; and (4) 5% of biopsies read as normal. Results will be classified for the study purposes as normal, low-grade (<CIN2), high-grade (CIN2,

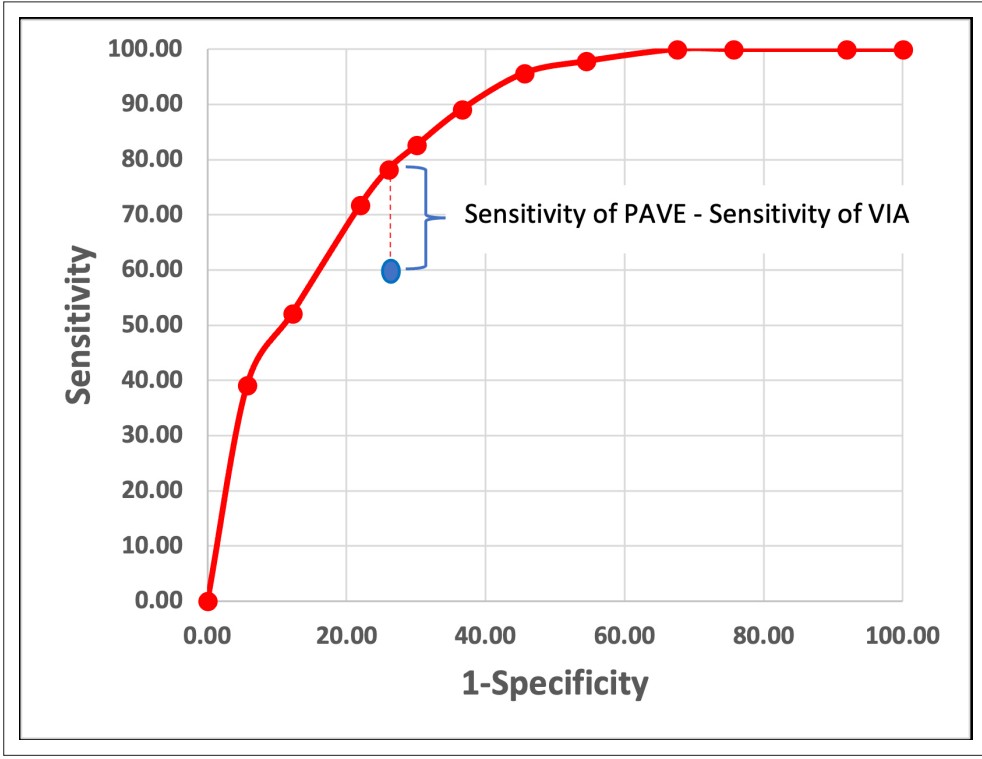

**Figure 6.** Theoretical approach to compare the PAVE strategy (which is HPV and AVE combined strategy. in red) and the standard of care (SOC) screening and triage outcome (which commonly is VIA, in blue). This figure represents a hypothetical example showing how testing for HPV and using AVE as triage (PAVE) and SOC will be compared. Under a specific specificity value (which will be determined by the SOC), we will compare the difference between the sensitivities of PAVE and SOC. In this example, we had three categories for human papillomavirus (HPV) genotype groups and three categories for automated visual evaluation (AVE) (*normal-indeterminate-precancer/cancer), and in total nine PAVE categories.* We estimate that about half of the CIN3 + cases will be positive for HPV16, and about 10–15% will be HPV18/45 positive, and that the remaining two other channels will have a 20% prevalence within cases.

**Table 2.** HPV-AVE risk strata.

| HPV risk group | AVE Risk Classification | | |
| --- | --- | --- | --- |
| | Precancer+ | Indeterminate | Normal |
| HPV16 | Highest | High | High |
| HPV18/45 | High | High | High |
| HPV31/33/35/52/58 | High | Medium | Medium |
| HPV39/51/56/59/68 | High | Medium | Low |
| Negative | Lowest | | |

CIN3), adenocarcinoma in situ (AIS), or invasive cervical cancer (squamous, adenosquamous, adeno-carcinoma, or other).

## Statistical analysis

The primary objective of the PAVE protocol is to compare the sensitivity at a given specificity level of the PAVE approach for triaging HPV-positive women to current SOC (HPV testing without genotyping triaged by VIA or colposcopic impression) (*Figure 6*). The PAVE screen-triage-treat protocol combines the three-class classification label (normal, indeterminate, precancer+) provided by the AVE algorithm with the four hrHPV risk group strata to create 12 strata of risk of precancer (*Table 2*).

The PAVE protocol will be compared to SOC VIA or colposcopy screen-triage protocols using the visual impressions recorded during the study. In SOC scenarios, participants are classified as positive or negative in the HPV test and as normal or abnormal in visual evaluations (e.g. VIA negative or positive, colposcopic impression less than high-grade or high-grade+) (*Table 3*).

The strata from *Table 2* will be plotted as a ROC-like curve of sensitivity versus (1-specificity). The ROC curve for each study site will yield an area under the curve (AUC) for both the PAVE and the SOC strategies.

The 12-stratum AVE risk score will be compared to a visual assessment of SOC: VIA (negative or positive) or colposcopic impression (less than high grade vs. high grade+) (*Egemen et al., 2023*). At each site, we will compare the sensitivity of the two approaches, at the (1-specificity) value produced by SOC visual evaluation. We will compile these values for the consortium and calculate the average (weighted by study size). We hypothesize that the difference of two sensitivities (PAVE minus SOC) conducted on the overall consortium data will be significantly greater than zero (the null hypothesis of no difference in sensitivity), indicating that PAVE detects more precancer than SOC triage at the same level of specificity. Where available, the analysis will be stratified by HIV status. Throughout the study, we will be checking for the consistency of results at the different steps across sites (e.g. performance of HPV results, quality of the images, histopathology reports). We will assess the reproducibility of the PAVE strategy across different sites by measuring the variability of the AUC values.

The list of variables collected in addition to HPV results and images is available on request.

Additional analyses that may be performed during the efficacy phase include:

- Development of treatability algorithm: In addition to defining SCJ visibility, an AI algorithm to assess whether a lesion fulfills WHO ablation criteria is in development. If AI algorithm development is successful, the output will be compared to the VIA or colposcopy assessment of eligibility for ablation or referral for surgical management. Accuracy of the three-class classification label provided by the AVE treatability algorithm (treatable, uncertain, not treatable) will be compared to the against a truth label based on the evaluation of three experts.

- Impact of HIV status on PAVE algorithm: We will evaluate the impact of HIV on the PAVE performance for the accuracy in the detection of precancer+ by comparing the accuracy in women

**Table 3.** Risk strata for participants with an HPV-positive women and visual Standard of Care (SOC) as the triage (i.e. VIA, colposcopy) test.

| HPV test result | SOC classification* | |
| --- | --- | --- |
| | Positive/High grade | Normal |
| Positive | High | Low |
| Negative | Lowest | |

Note: Participants with a negative test for HPV will not have a VIA nor colposcopy assessment. see *Table 1* for SOC at individual PAVE sites.

with and without HIV, with some consideration of the role of effective ART (*Parham et al., 2023*).

## Phase 2: Effectiveness

During the efficacy phase, the PAVE algorithm is undergoing evaluation and development, and clinicians will not be provided with HPV genotyping, AI algorithm outputs, or risk strata. When the efficacy phase is complete, if the PAVE algorithm outperforms the SOC, we will begin the effectiveness phase. Sites that chose to participate in the effectiveness phase will test the following screen-treat-triage protocol. Screening: self-sampled HPV testing with genotyping (ScreenFire or equivalent test). Triage of HPV-positive individuals: image collection using the Iris device, upon which the AI algorithms have been installed. The AI algorithms will guide the clinician in taking high-quality images (cervix identifier, image quality classifier) and then provide a disease classification score of normal, indeterminate, or precancer+ and an SCJ visibility assessment of fully visible, partially visible, or not visible. The clinician will then enter the HPV genotyping test result and the device will output a risk category using the strata in *Table 2* (lowest, low, medium, high, highest). The risk category and SCJ visibility assessment are designed as clinical management tools to aid clinicians in determining which patients are most likely to benefit from treatment, and among those needing treatment, whether ablation can be considered. This phase will assess the feasibility and acceptability of the PAVE strategy in clinical practice.

### Cost-effectiveness analysis

To inform decision makers designing cervical cancer prevention programs in resource-limited settings, we will analyze the cost-effectiveness (i.e. cost per precancer detected) and affordability (the impact on a payer's budget) of the PAVE strategy at several sites. Micro-costing efforts to estimate the cost of all resource ingredients used for a screening episode are underway with technical assistance provided to study sites. A microsimulation model of genotype-specific HPV natural history and cervical carcinogenesis is being developed specifically for evaluation of novel biomarker triage tests, including AVE (*Campos et al., 2021*). By adapting this natural history model to setting-specific HPV prevalence patterns (by genotype and age); overlaying screening, triage, and treatment strategies; and using setting-specific healthcare delivery data inputs for uptake, adherence to management, and costs, we will evaluate the cost per precancer detected by the PAVE strategy relative to SOC. We will explore the health and economic implications of applying different management approaches (e.g. triage, treatment, and follow-up) to different risk strata, depending on health system capacity.

The development of the microsimulation model and the micro-costing tools for the PAVE consortium will serve as the basis for estimating the real-world costs and health benefits of implementing novel screening and management strategies. These tools can be adapted to different settings, with refinement of management algorithms, health care delivery variables, and cost estimates as implementation and scale-up occur. An early exercise to approximate the potential costs and benefits of a highly effective screening campaign delivered to women aged 30–49 years in the ~65 highest burden LMIC (*Ferlay et al., 2020*; *GAVI, 2023*; *Akumbom et al., 2022*; *World Bank, 2023*; *Figure 1*) and an HPV vaccination program delivered to girls ages 9–14 years found that the number of screening or adolescent HPV vaccinations needed to avert one cervical cancer death was similar for each intervention (i.e. 293 for screening; 278 for vaccination). Assuming a bundled cost of US$15 per woman screened and managed appropriately, a one-time screening campaign that achieves 40% uptake and ~60% adherence to recommended treatment for screen-positive women yielded a financial cost of ~US$2.5 billion to avert ~570,000 deaths, or US$4400 per death averted. On a similar order of magnitude, a one-time single-dose bivalent HPV vaccination campaign achieving 90% coverage of girls aged 9–14 years in the same countries (US$4.50 vaccine cost; US$7 delivery cost) would cost ~US$2.0 billion and avert ~640,000 deaths, or US$3200 per death averted. Of note, these ballpark estimates are undiscounted and do not account for cancer treatment cost offsets. While data are not yet available on the costs of implementing novel highly effective screening strategies for adult women and single-dose HPV vaccination for female adolescents, these data are forthcoming from the PAVE consortium and single-dose vaccination studies. Refining these cost and effectiveness estimates, and obtaining country-specific data, is a high priority and a critical component of the PAVE consortium objectives.

## Stakeholder knowledge and attitudes regarding cervical cancer prevention and screening interventions in the PAVE study

Effective dissemination and implementation of the PAVE strategy in the future will require clear and consistent strategies for communicating information to healthcare providers and patients. The field of HPV and screening is rapidly evolving and constantly being enriched with new scientific information. However, this influx of information can sometimes lead to unclear or conflicting messages, which in turn may diminish the effectiveness of interventions aimed at improving screening rates and optimizing management strategies. Furthermore, both healthcare providers and patients can differ in their knowledge and perceptions of cervical cancer risk, tolerance of risk, and their personal values and attitudes regarding cervical cancer prevention and screening, all of which can influence the uptake of prevention strategies such as the PAVE strategy. To address these challenges, it will be necessary to provide healthcare providers with accurate, consistent information about the latest scientific advances and guidelines, and as well as training and tools that can help them effectively inform and engage patients in cervical cancer prevention and screening.

To address these needs and prepare for broader dissemination and implementation of the PAVE strategy, the Communication and Retention Workgroup is conducting a mixed-methods pilot study utilizing both qualitative interviews and survey questionnaires administered to scientific experts as well as key stakeholders—patients and healthcare providers—at four participating sites (Brazil, El Salvador, Nigeria, Tanzania). The specific objectives of this pilot study are to explore stakeholders' knowledge, perceptions, and attitudes regarding cervical cancer prevention and screening, and their preferences for information and participation in decision-making. The study will generate evidence to enable the future development of effective, ethical strategies for engaging eligible members of LMIC communities in cervical cancer prevention and screening, including using the PAVE strategy.

## Discussion

### Timeline and next steps

At the time of this writing, the study has been initiated in nine countries. It is expected that a preliminary interim analysis on the efficacy of the PAVE strategy will be completed by the early 2024, and field recruitment will be completed by end of 2024. The efficacy phase is designed to assess the validity of the PAVE protocol by refining the AVE protocol using histopathology specimens and demonstrate the superiority of PAVE for detecting precancer and minimizing unnecessary referrals compared to

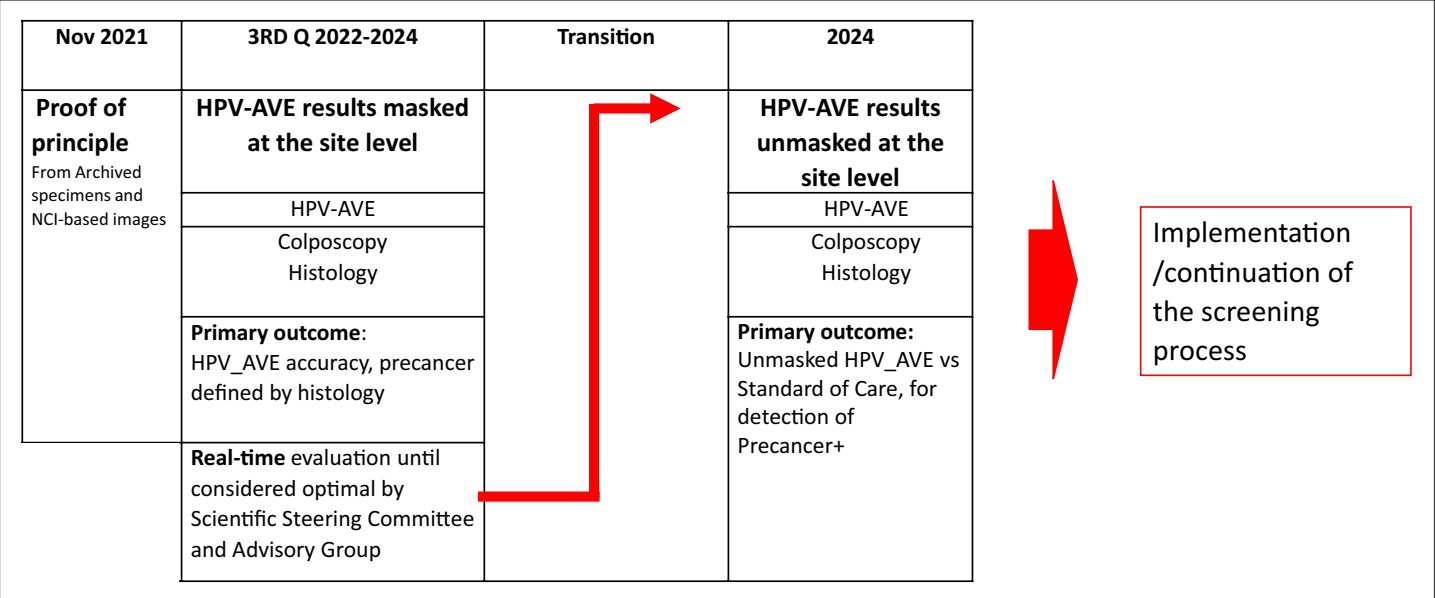

**Figure 7.** Estimated timeline of HPV-automated visual evaluation (PAVE) validation.

the SOC. Following the initial efficacy phase, the effectiveness phase is planned to examine additional factors including program feasibility, acceptability, and cost-effectiveness.

In the *Figure 7*, a schematic timeline of the project from proof of principle to the estimated final efficacy and effectiveness steps.

## Regulatory considerations

The rapid advancement of AI in healthcare has prompted significant ethical and regulatory discussions. Global ethical principles specific to AI in healthcare are in development and are rapidly evolving, regulatory authorities such as the WHO and the U.S. Food and Drug Administration (FDA) are responsible for developing guidelines that ensure the safe, effective, and appropriate use of AI technologies in healthcare and therapeutic development. In addition to complying with existing ethical principles in medicine, AI solutions must demonstrate scientific validity, ensuring their effectiveness and reliability. It is imperative that AI technologies do not perpetuate discrimination or bias, or exclude specific population segments. One major concern is the potential creation of a permanent digital identity, linked to individuals' health and personal data, without obtaining proper consent (*WHO, 2021*, *FDA, 2023*).

The Iris device, used in PAVE, is an example of an AI-based system in healthcare. It will incorporate AI algorithms that aim to generate scores evaluating image quality, presence of precancerous lesions, and visibility of the SCJ based on digital cervix images. These scores provide additional information to assist healthcare providers in making informed decisions. To arrive at a comprehensive decision, the provider will need to integrate the HPV information, gynecological examination findings, scoring system outputs generated from the AI algorithms, and the patient's medical history. Considering its functionality, the AI algorithm used in the PAVE system, based on FDA specifications, can be considered an assistant to medical management offering the joint evaluation of the HPV data and the AVE outcome. As WHO is actively evaluating screening and triage approaches, it is expected that clearer regulatory guidance on the use of the algorithms will be available before their implementation (*WHO, 2021*; *FDA, 2023*).

## Competition and commercial aspects

The PAVE strategy utilizes certain products of significant commercial importance. The selection of each device or assay is based on factors such as accuracy to achieve its purpose (i.e. high sensitivity of the ScreenFire HPV test to detect precancer, high-quality image capture by the Iris device) and affordability. It is important to note that NCI PAVE collaborators have no commercial interests that could create appearance of conflict of interest. But for collaborators does not preclude future commercial developments. The presence of competition with similar assays is desirable as long as they can provide similar qualities by commercial partners. If the PAVE strategy is proven to be more accurate than the SOC for screening and triage, there will likely be an increased demand for devices and assays. Procurement of these resources may present challenges and require effective communication with the respective Ministries of Health to ensure a smooth introduction in countries where the need is evident.

## Dissemination

After the demonstration of efficacy and effectiveness, we expect to have a screen-treat-triage protocol fulfilling WHO principles that can be used at the discretion of local health authorities in LMIC for cervical cancer prevention programs. If the PAVE strategy is proven to outperform the local SOC, and be feasible, acceptable, and affordable for resource-limited settings, then countries may switch their current SOC to the PAVE strategy. De-implementation of existing strategies, such as cytology, colposcopy, and VIA, as well as implementation of self-sampled HPV with AVE triage will require buy-in from stakeholders and policymakers as well as substantial investment in educating and retraining the laboratory and clinical workforces. Cytology/colposcopy programs, though effective, are limited in scope and are costly to maintain, therefore, switching may be attractive to health ministers. However, laboratories will need funding for the purchase of equipment for running HPV testing and materials for self-sampling, and the cytotechnologist and pathologist workforce will be reduced. Countries with existing VIA programs will require significant introduction costs such as laboratory machinery and training of healthcare personnel, and recurrent costs including reagents and self-sampling materials.

In all settings, program continuation beyond the initial study will require local governments and programs to address issues of procurement and implementation, as well as de-implementation of existing strategies.

In conclusion, the PAVE project will develop and validate a strategy using self-sampled HPV with genotyping and AVE to identify precancer in a large group of women from many different settings. The PAVE objective is to create an accurate, feasible, cost-effective screening and triage protocol for cervical cancer prevention in resource-limited settings. If proven effective, cost-effective, feasible, and acceptable, the strategy can have a major impact in reducing cervical cancer particularly among non-vaccinated adult women.

## Acknowledgements

Farideh Almani for secreterial support.

## Additional information

### Competing interests

Brian Befano: is an employee of Information Management Services Inc. The other authors declare that no competing interests exist.

### Funding

| Funder | Grant reference number | Author |
| --- | --- | --- |
| Intramural Research Program of the National Cancer Institute | | Silvia de Sanjosé Rebecca B Perkins Nicole Campos Federica Inturrisi Didem Egemen Brian Befano Ana Cecilia Rodriguez Jose Jerónimo Li C Cheung Kanan Desai Paul Han Abigail Ukwuani Syed Rakin Ahmed Nicolas Wentzensen Mark Schiffman |

The funders had no role in study design, data collection and interpretation, or the decision to submit the work for publication.

### Author contributions

Silvia de Sanjosé, Conceptualization, Data curation, Investigation, Methodology, Supervision, Validation, Writing – original draft, Writing – review and editing; Rebecca B Perkins, Writing – review and editing, Methodology, Conceptualization, Writing – original draft; Nicole Campos, Validation, Formal analysis, Methodology, Conceptualization, Writing – original draft; Federica Inturrisi, Data curation, Investigation, Conceptualization, Writing – original draft; Didem Egemen, Ana Cecilia Rodriguez, Formal analysis, Methodology, Conceptualization, Writing – original draft; Brian Befano, Formal analysis, Investigation, Methodology, Writing – original draft; Jose Jerónimo, Methodology, Writing – original draft; Li C Cheung, Formal analysis, Data curation, Writing – original draft; Kanan Desai, Data curation, Methodology, Conceptualization, Writing – original draft; Paul Han, Validation, Data curation, Methodology, Writing – original draft; Akiva P Novetsky, Writing – review and editing, Data curation, Writing – original draft; Abigail Ukwuani, Project administration, Writing – original draft; Jenna Marcus, Writing – review and editing, Writing – original draft; Syed Rakin Ahmed, Software, Formal analysis, Methodology, Conceptualization, Writing – original draft; Nicolas Wentzensen, Methodology, Conceptualization, Writing – original draft; Jayashree Kalpathy-Cramer, Validation, Formal analysis, Data curation, Investigation, Writing – original draft; Mark Schiffman, Validation, Data curation, Methodology, Conceptualization, Supervision

## Author ORCIDs
Silvia de Sanjosé [ID] https://orcid.org/0000-0002-5909-676X
Federica Inturrisi [ID] http://orcid.org/0000-0002-3661-0842
Li C Cheung [ID] http://orcid.org/0000-0003-1625-4331
Kanan Desai [ID] http://orcid.org/0000-0002-8992-5944
Syed Rakin Ahmed [ID] http://orcid.org/0000-0002-1615-8633
Mark Schiffman [ID] http://orcid.org/0000-0002-4625-2508

## Ethics
Informed consent has been obtained in each study site together with the corresponding ethical approval at each site.

Reviewer #1 (Public Review): https://doi.org/10.7554/eLife.91469.3.sa1
Reviewer #2 (Public Review): https://doi.org/10.7554/eLife.91469.3.sa2
Reviewer #3 (Public Review): https://doi.org/10.7554/eLife.91469.3.sa3
Author Response https://doi.org/10.7554/eLife.91469.3.sa4

---

# Additional files

## Supplementary files
• MDAR checklist

## Data availability
In the Protocol phase of the PAVE study we do not report neither previously generated nor newly generated datasets. There is no code/software/algorithm available to be shared.

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

## Appenidx 1

### PAVE Study Group

#### Brazil

Ana Ribeiro - ana-ribeiro.dantas@fiocruz.br, Tainá Raiol - taina.raiol@fiocruz.br, Center for Women's Integrated Health, Oswaldo Cruz Foundation (Fiocruz), Brasília, DF, Brazil. MARCO Clinical and Molecular Research Center, University Hospital of Brasília/EBSERH, Federal District, Brazil

#### Cambodia

Te Vantha, MD, Director of Takeo Provincial Hospital, Cambodia. Thay Sovannara, MD, Medical Practitioner, Raffles Medical Group, Cambodia. Judith Norman, MD, Director of Women's Health, Mercy Medical Center, Cambodia, judynorman@gmail.com. Dr. Andrew T. Goldstein, Director, Gynecologic Cancers Research Foundation drg.cvvd@gmail.com.

#### Dominican Republic

Margaret M. Madeleine, MPH, PhD, Program in Epidemiology, Fred Hutchinson Cancer Centermmadelei@fredhutch.org. Yeycy Donastorg, MD, Instituto Dermatológico y Cirugía de la Piel "Dr. Huberto Bogaert Díaz", HIV Vaccine Trials Research Unit, Santo Domingo, Dominican Republic. ydonastorg@gmail.com

#### El Salvador

Miriam Cremer MD; Basic Health International, Pittsburgh, PA 15205, USA. Ob/Gyn and Women's Health Institute, Cleveland Clinic, Cleveland, OH 44195, USA miriam.cremer@gmail.com. Karla Alfaro, MD Basic Health International, El Salvador, kalfaro@basichealth.org

#### Honduras

Miriam Cremer MD; Basic Health International, Pittsburgh, PA 15205, USA. Ob/Gyn and Women's Health Institute, Cleveland Clinic, Cleveland, OH 44195, USA, miriam.cremer@gmail.com. Karla Alfaro, MD Basic Health International, El Salvador, kalfaro@basichealth.org. Jaqueline Figueroa, MD, Programa Nacional contra el Cáncer, Tegucigalpa, Honduras. jacqueline_figueroan@yahoo.com

#### Eswatini

Eyrun F. Kjetland, MD, PhD, Professor, Departments of Global Health and Infectious Diseases Ullevaal, Centre for imported and Tropical Diseases, Oslo University Hospital Ullevaal, Oslo, Norway; College of Health Sciences, Discipline of Public Health, Nelson Mandela School of Medicine, University of KwaZulu-Natal, Durban, South Africa; Centre for Bilharzia and Tropical Health Research (non-profit), BRIGHT Academy, Durban, South Africa e.f.kjetland@medisin.uio.no, Teresa Norris, Founder and President, HPV Global Action, tnorris@hpvglobalaction.org. Zeev Rosberger, PhD, Department of Oncology, Psychology and Psychiatry, McGill University, Montreal, Canada, zeev.rosberger@mcgill.ca. Amelie McFadyen, MA, Chief Executive Officer, HPV Global Action, ameliemcfadyen@hpvglobalaction.org. Marc Steben, MD, Ecole de Sante Publique, Université de Montréal; International society for STD research, marc@marcsteben.com

#### Malawi

Amna Haider, MD, Epidemiologist, Department of Epidemiology and Training, Epicentre, Dubai, UAE, amna.haider@epicentre.msf.org. George Kassim Chilinda, MD, Médecins Sans Frontières, Operational Centre Paris, Blantyre, Malawi, gchilinda@gmail.com. Henry B.K.Phiri, MD- Sexual and reproductive health department, Ministry of Health, Malawi, henryphiri06@gmail.com

#### Nigeria

Ajenifuja Kayode Olusegun, MD, Obafemi Awolowo University Teaching Hospital, Ile-Ife, Osun state Nigeria, ajenifujako@yahoo.com. Adepiti Clement Akinfolarin, MD, Obafemi Awolowo University Teaching Hospital, Ile-Ife, Osun state Nigeria, akinfolarindepiti@yahoo.co.uk. Adekunbiola Banjo, MD, College of Medicine University of Lagos, Lagos, aafbanjo@cmul.edu.ng. Moharson-Bello Imran, MD, College of Medicine, University of Ibadan, Oyo state, Nigeria, imranmorhasonbello@gmail.com. Oyinloye Temitope, MD, Obafemi Awolowo University Teaching Hospitals Complex, Ile-Ife,

Osun state, Nigeria, projectcoordinator.itoju@gmail.com. Bola-Oyebamiji Sekinat, MD, College of Medicine, Osun state University, Osogbo, Osun state. Adeyemo Marydiya, MD, College of Medicine, Osun state University, Osogbo, Osun state

## Tanzania

Karen Yeates-MD, MPH, Department of Medicine, Queen's University, Kingston, Ontario, Canada, yeatesk@queensu.ca. Safina Yuma, MD, Cervical Cancer Focal Person, Ministry of Health, Tanzania, sychande@yahoo.com. Bariki Mchome, MD, Head, Reproductive Health Centre, Kilimanjaro Christian Medical Centre, Kilimanjaro, Tanzania, barikimchome@gmail.com. Alex Mremi, MD, Head, Department of Pathology, Kilimanjaro Christian Medical Centre, Kilimanjaro, Tanzania, alexmremi@gmail.com.

