## [Editor Report · eLife assessment]

This **important** study will provide evidence about a novel screen-triage-treat strategy for cervical cancer prevention. The trial will generate **convincing** evidence regarding the efficacy, effectiveness, cost-effectiveness, feasibility and acceptability in a range of geographically spread low-resource settings. The strategy should contribute to improving access to cervical cancer prevention to vulnerable women with low access to health care, and, therefore, at the highest risk of cervical cancer.

---

## [Referee Report · Reviewer #1 (Public Review)]

Summary: A description of a modern protocol for cervical screening that likely could be used in any country of the world, based on self-sampling, extended HPV genotypinng and AI-assisted visual inspection - which is probably the best available combination today.

Strengths: Modern, optimised protocol, designed for global use. Innovative.

Weaknesses: The protocol is not clear. I could not even find how many women were going to be enrolled, the timelines of the study, the statistical methods ("comparing" is not statistics) or the power calculations.

Tables 2 and 3 are too schematic - surely the authors must have an approximate idea of what the actual numbers are behind the green, red and yellow colors.

Figure 1 comparing screening and vaccination is somewhat misleading. They screen 20 birth cohorts but vaccinate only 5 birth cohorts. Furthermore, the theoretical gains of screening has not really been attained in any country in practise. Modelling can be a difficult task and the commentary does not provide any detail on how to evaluate what was done. It just seems unnecessary to attack vaccination as a motivation on why screening needs to be modernised.

---

## [Referee Report · Reviewer #2 (Public Review)]

Summary:

This manuscript describes the study protocol, structure and logic of the PAVE strategy. The PAVE study is a multicentric study to evaluate a novel cervical screen-triage-treat strategy for resource-limited settings as part of a global strategy to reduce cervical cancer burden. The PAVE strategy involves: (1) screening with self-sampled HPV testing; (2) triage of HPV-positive participants with a combination of extended genotyping and visual evaluation of the cervix assisted by deep-learning-based automated visual evaluation (AVE); and (3) treatment with thermal ablation or excision (Large Loop Excision of the Transformation Zone). The PAVE study has two phases: efficacy (2023-2024) and effectiveness (planned to begin in 2024-2025). The efficacy phase aims to refine and validate the screen-triage portion of the protocol. The effectiveness phase will examine few implementation of the PAVE strategy into clinical practice. In following phases implementation will further explored.

Strengths and weaknesses

The Pave Study develops and evaluates a novel strategy that combines HPV self-collection -that has been proven effective to increase screening coverage in different settings-, with genotyping and Automated Visual Evaluation as triage. The proposed strategy combined three key innovations to improve an important step in the cervical cancer care continuum. If the strategy is effective it will contribute to enhance cervical cancer prevention in low resource settings.

As authors mentioned, despite the existence of effective preventive technologies (e.g., HPV vaccine and HPV test) translation of the HPV prevention methods has not yet occurred in many Low-Middle-Income Countries. So, in this context, new screen-triage-treat strategies are needed and if PAVE strategy were effective, it could be a landmark for cervical cancer prevention.

The PAVE Study is a solid and important study that is aimed to be carried out in nine countries and recruit tens thousands of women. It is a study with a large and diverse sample that can provide useful information for the development of this new screen-triage-treat strategy. Another strength is the fact that the PAVE project is integrated into the screening activities placed in the selected countries that will allow to evaluate efficacy and effectiveness in real-word context.

The manuscript does not present results because its aim is to describe the study protocol, structure and logic of the PAVE strategy.

Phase 1 aims to evaluate efficacy of the strategy. Methods are well described and are consistent with the study aims.

Phase 2 aims to evaluate the implementation of the PAVE strategy in clinical practice. The inclusion of implementation evaluation in this type of studies is an important milestone in the field of cervical cancer prevention. It has been shown that many strategies that have proven to be effective in controlled studies face barriers when they are implemented in real life. In that sense, results of phase 2 are key to ensure the future implementation of the strategy.

---

## [Referee Report · Reviewer #3 (Public Review)]

Summary: Despite being preventable and treatable, cervical cancer remains the second most common cause of cancer death globally. This cancer, and associated deaths, occur overwhelmingly in low- and middle-income countries (LMIC), reflecting a lack of access to vaccination, screening and treatment services. Cervical screening is the second pillar in the WHO strategy to eliminate cervical cancer as a public health problem and will be critical in delivering early gains in cervical cancer prevention as the impact of vaccination will not be realized for several decades. However, screening strategies implemented in high income countries are not feasible or affordable in LMICs. This ambitious multi-center study aims to address these issues by developing and systematically evaluating a novel approach to cervical screening. The approach, based on primary screening with self-collected specimens for HPV testing, is focused on optimizing triage of people in whom HPV is detected, so that sensitivity for the detection of pre-cancer and cancer is maximized while treatment of people without pre-cancer or cancer is minimized.

Strengths:

The triage proposed for this study builds on the authors' previously published work in designing the ScreenFire test to appropriately group the 13 detected genotypes into four channels and to develop automated visual evaluation (AVE) of images of the cervix, taken by health workers.

The move from mobile telephone devices to a dedicated device to acquire and evaluate images, overcomes challenges previously encountered whereby updates of mobile phone models required retraining of the AVE algorithm.

The separation of the study into two phases, an efficacy phase in which screen positive people will be triaged and treated according to local standard of care and the performance of AVE will be evaluated against biopsy outcomes will be followed by the second phase in which the effectiveness, cost-effectiveness, feasibility and acceptability will be evaluated.

The setting in a range of low resource settings which are geographically well spread and reflective of where the global cancer burden is highest.

Weaknesses:

Potential ascertainment bias due to the lack of specified biopsy (such as small four quadrant biopsies or small biopsies across the transformation zone) when aceto-white areas are not identified. This has the potential lead to lead to an over-estimate of sensitivity of the triage approach, particularly in the setting of VIA as compared to colposcopy. While the authors specify endocervical sampling in this setting, using curette or brush (for cytology), this may not be as sensitive unless clinicians are experienced in endocervical curette procedures.

---

## [Author Response]

The following is the authors’ response to the original reviews.

**Reviewer #1 (Recommendations For The Authors):**
It is not clear if the cost-effectiveness cited refers exactly to the PAVE protocol. No line item costings are given. As far as I know, the AmpFire test is very expensive (some 6 USD) and AI-assisted colposcopy has at least formerly been very expensive.

Response: As mentioned in the section on "Cost-effectiveness analysis," the cost-effectiveness results refer to "an early exercise to approximate the potential costs and benefits of a highly effective screening campaign delivered to women aged 30-49 years in the ~65 highest burden LMIC (Figure 1; Suppl Materials) and an HPV vaccination program delivered to girls aged 9-14 years". Because this modeling was intended to be a high-level approximation prior to the availability of micro-costing and use of a new microsimulation model reflecting the epidemiology of HPV in PAVE study sites, we used a bundled cost of US$15 per woman screened and managed appropriately, including the ~$6 cost of the ScreenFire test, triage with AVE for women with HPV positivity, and treatment based on risk stratification. Micro-costing and microsimulation model development for PAVE sites are ongoing alongside the study and will have the capability to reflect setting-specific differences in delivery costs, as well as different burdens of HPV and precancer. These refinements of costing and cost-effectiveness estimates are a high priority of the PAVE consortium

**Reviewer #2 (Recommendations For The Authors):**
As mentioned above, the description of phase 2 could be improved. I suggest that the inclusion of Implementation Science frameworks and tools could contribute to strengthening methods to measure implementation outcomes. Perhaps if the protocol and scope of the study allows it, I suggest that the authors evaluate the incorporation of the assessment of barriers and facilitators of implementation to inform future scaling up of the PAVE strategy. To do this, for example, some Implementation Science Frameworks, such as Conceptual Framework of Implementation Research (CFIR)1-2 could be useful. In addition, as the authors mentioned, future dissemination will need an effective communication strategy and to design it they will carry out a pilot study. The inclusion of CFIR framework or other similar framework, could contribute to identifying contextual factors that might affect implementation and contribute to designing an accurate implementation and dissemination strategy.The authors also mentioned that if the PAVE strategy is effective, it could replace the current standard of care. This fact would lead to the need to carry out a des-implementation process. This process needs stakeholders' engagement and political will, among other contextual factors (e.g., human resources, organizational changes, etc.).Implementation of new strategies needs that implementers perceive it as acceptable, adaptable, compatible and with greater advantages than the usual practice. In this sense, the analysis of implementation outcomes guided by CFIR framework could play an important role in this future des-implementation process.1. Damschroder, et al. Fostering implementation of health services research findings into practice: a consolidated framework for advancing implementation science. Implementation Sci 4, 50 (2009) https://doi.org/10.1186/1748-5908-4-50.1. Damschroder, L.J., Reardon, C.M., Widerquist, M.A.O. et al. The updated Consolidated Framework for Implementation Research based on user feedback. Implementation Sci 17, 75 (2022). https://doi.org/10.1186/s13012-022-01245-0

Response: Phase 2 refers to limited aspects of PAVE implementation, mainly introducing the management algorithms and evaluating the acceptability by providers and patients. Based on preliminary results of PAVE in the efficacy analysis a more comprehensive implementation intervention is being planned.

**Reviewer #3 (Recommendations For The Authors):**
This is a very strong protocol and obviously the synthesis of many years' of work. I have some minor suggestions only.The issue raised as a weakness could be addressed by specifying that biopsy adequacy is evaluated by the local histopathologist. Those cases that don't contain at least some stroma and only superficial strips of epithelium should probably be assessed as "unsatisfactory" and excluded from triage performance calculations.While endocervical curettage is commonly performed in North America, resulting in good quality samples, there is considerable global variation in this practice. The procedure yielding high quality samples is usually somewhat painful due to the cervical dilation and may in fact be more painful than small biopsies.

Response: We are undertaking a thorough evaluation of histology assessment together with the on-site pathologists and an external expert reviewer. It is critical that the study material be of good quality and that the diagnosis be highly accurate as these elements are critical for patient management but also for an adequate training of the AI algorithm. We are recommending to use for endocervical sampling a soft tissue by Histologics that provides excellent material and it is reported to be less painful than regular curette. Pathologists are requested to verify the quality of the sampling of this approach.

The sentence starting at line 311 could add that, clinicians also record transformation type and/ or colposcopy adequacy.

Response: Added

The clinicians are reporting the VIA or the colposcopy impression and also the visibility of the SCJ.The manuscript could be strengthened by specifying what will happen to people who have HPV detected and are triage negative. Will they be recalled for follow-up HPV test at around 12 months or some other interval?Finally, will those who have been treated be recalled for a follow-up HPV test at around 12 months, particularly those treated with thermal ablation? Follow-up of people in whom HPV is detected, whether triage negative or positive (and treated) would strengthen the study and enhance participant safety. If this is already planned it would strengthen the manuscript to cover these aspects.

Response: The PAVE strategy runs under a Consortium agreement and thus we cannot dictate specific protocols for follow-up. We are very eager to promote an adequate follow-up for those with a triage test negative, but the monitoring of its implementation is beyond PAVE. All settings have under their guidelines a yearly follow-up for any woman receiving thermal ablation and shorter intervals for those getting LEEP (LLETZ).